# Nickel-Modified TS-1 Catalyzed the Ammoximation of Methyl Ethyl Ketone

**Dandan Yang, Haiyan Wang, Wenhua Wang, Sihua Peng, Xiuzhen Yang, Xingliang Xu \* and Shouhua Jia \***

College of Chemistry and Material Science, Shandong Agricultural University, Tai'an 271018, China;
2017110585@sdau.edu.cn (D.Y.); why15053865003@163.com (H.W.); w17863807408@163.com (W.W.);
psh18865383970@163.com (S.P.); x976360872@163.com (X.Y.)
**\*** Correspondence: xxlsdau@163.com (X.X.); shjia@sdau.edu.cn (S.J.); Tel.: +86-0538-824-1570 (S.J.)

**Abstract:** In this paper, five kinds of transition metal-modified titanium silicalite-1 (M-TS-1) were prepared by an ultrasonic impregnation method. We studied their catalytic performances in the ammoximation of methyl ethyl ketone (MEK). The various M-TS-1 catalysts revealed distinct differences in their MEK ammoximation activity. The nickel-modified TS-1 (Ni-TS-1), especially 3 wt % Ni-TS-1, exhibited a satisfactory conversion of MEK (99%) associated with a high selectivity of methyl ethyl ketoxime (MEKO) (99.3%), which was 6% higher than that of TS-1 under the same conditions. Moreover, the catalyst showed excellent recyclability and the reactivity could be completely recovered after regeneration. The catalysts were characterized by Powder X-ray Diffraction (XRD), Fourier Transformed Infrared Spectra (FT-IR), X-ray photoelectron spectroscopy (XPS), and so on. It was demonstrated that the skeleton structure of TS-1 was basically maintained and the electron environment of the Ti active site was changed after the nickel modification, which can optimize the adsorption capacity and the activation for $H_2O_2$. Meanwhile, the surface nickel species reduced the surface acidity of the catalyst, which provided an appropriate pH and inhibited the deep oxidation of oxime.

**Keywords:** TS-1; nickel-modified; methyl ethyl ketone; ammoximation; methyl ethyl ketoxime

## 1. Introduction

Oxime is an important chemical raw material and intermediate, which is widely used in the synthesis of various high value-added chemicals. For example, cyclohexanone oxime is a key intermediate in the production of caprolactam as nylon-6 monomer [1], and methyl ethyl ketoxime (MEKO) can also be used as an important raw material to synthesize silicone crosslinkers, silicon curing agents, and the blocking agents of isocyanate, etc. [2–4]. The traditional synthesis method of oxime is the hydroxylamine method, which is a convenient and valuable method and a non-catalytic oximation of ketone with hydroxylamine derivative like $(NH_2OH)_2·H_2SO_4$. Unfortunately, the hydroxylamine method has many drawbacks, such as using toxic and highly acidic reagents, like hydroxylamine and sulfuric acid, while producing a large number of low-value by-products such as ammonium sulfate [5]. Compared with the traditional hydroxylamine method, the ammoximation with titanium silicate molecular sieve as catalyst and $H_2O_2$ as oxidant is a new method for the preparation of ketoxime which has been developed in recent years. Based on the concept of "green chemistry", the method has a series of advantages such as high efficiency, mild reaction conditions, atom economy, and only water as by-product (Scheme 1) [6,7].

Since Taramasso first synthesized titanium silicalite-1 (TS-1) [8], the development of titanium silicalite molecular sieve as a catalyst has been widely concerned. The active center of TS-1 with

MFI topology is $Ti^{4+}$ in the framework of molecular sieve. Because titanium oxygen tetrahedron is unstable, it is difficult to exist in the perfect form of tetra-coordination. It has the tendency to form six-coordination [9]. This means that the tetra-coordinated $Ti^{4+}$ has electronic defects and the potential of accepting electron pairs. Therefore, it has unique adsorption and activation properties for $H_2O_2$ and catalyzes the selective oxidation of a variety of organic compounds [10]. At present, the ammoximation of cyclohexanone catalyzed by TS-1 has been industrialized. The conversion of cyclohexanone reached 99.9%, and the selectivity of cyclohexanone oxime was higher than 98.2% [11]. However, when the TS-1/$H_2O_2$ system was applied to the ammoximation of small ketones like methyl ethyl ketone (MEK), the selectivity of MEKO was not satisfactory. Under the same reaction conditions with cyclohexanone, the selectivity of MEKO is only 80%. The selectivity of MEKO can only reach about 95% by optimizing the conditions. The reason may be that the linear small ketoxime is easy to enter into the pore of the catalyst and is oxidized deeply [4].

$$R_1{=}O \;+\; NH_3 \;+\; H_2O_2 \;\xrightarrow{\text{Catalyst}}\; R_1{=}NOH \;+\; 2H_2O$$

**Scheme 1.** Preparation of oxime by the ammoximation of ketone.

Regulating electrophilicity of TS-1 to optimize adsorption ability and the Lewis acidic strength is a very effective strategy for preparing high-efficiency TS-1-based catalysts and improving the applications of TS-1 in oxidation reaction [12–14]. According to the catalytic properties of TS-1 and the mechanism of ammoximation [15], the modification of TS-1 to regulate its electrophilicity may be important to improve the selectivity of ketoxime in the ammoximation of ketone.

In recent years, transition metals (such as Ni, Fe, Co, Cu, etc.)-based catalysts have been extensively studied and applied in catalytic oxidation [16–19]. Some transition metal-modified TS-1 were used in the epoxidation of olefin. Wu et al. investigated the effect of transition metal (such as V, Cr, Mn, Fe, Co, Ni, Cu, Zn, Cd, La, 1% metals loading)-modified TS-1 on the epoxidation of butadiene. They found that Fe, Co, Ni can promote $H_2O_2$ conversion effectively and increase the electrophilicity of TS-1. These catalysts exhibit significant enhancement in butadiene selective epoxidation. However, Cu can inhibit $H_2O_2$ conversion in this reaction, and the interaction between Cu and Ti is relatively weak [20,21]. Capel-Sanchez et al. carried out the epoxidation of 1-octene with the TS-1 modified by several metal ions ($Li^+$, $Ca^{2+}$, $La^{3+}$, and $Ce^{4+}$). They found that the addition of metal cations could significantly improve the selectivity of epoxides, which is attributed to the addition of metal oxides neutralizing the surface acidity of TS-1 zeolite, and thus inhibiting the solvolysis reaction of epoxides at acidic sites and improving the selectivity of epoxides [22]. Nevertheless, there are few reports about transition metal-modified TS-1 used in the ammoximation of ketone.

Herein, we prepared a series of transition metal-modified TS-1 catalysts by an ultrasonic impregnation method. In comparison with the existing catalysts, the nickel-modified TS-1 catalysts, especially 3 wt % Ni-TS-1, showed great improvements in the ammoximation of MEK to synthesize MEKO with $H_2O_2$ as the oxidant. A detailed characterization and the effects of various experimental parameters were systematically conducted and investigated, where the nature of the promoted catalytic performances in the ammoximation of MEK was revealed. Moreover, the catalyst has successfully recovered without considerable loss of MEK conversion and MEKO selectivity.

## 2. Results and Discussion

### 2.1. Catalytic Activity on the Ammoximation of MEK

#### 2.1.1. The Catalytic Activity of M-TS-1

Five kinds of transition metal (Fe, Co, Ni, Cu, Ce)-modified TS-1 catalysts were prepared. Moreover, the catalytic performances of M-TS-1 on the ammoximation of MEK were studied. The results are shown in Table 1. It is apparent that the various M-TS-1 catalysts revealed distinct different catalytic

activity. The catalytic effect of Cu-TS-1 was the worst. The conversion of MEK was reduced from 98.9% of TS-1 to 74.1%. The selectivity of MEKO was 0, that is, MEK was all converted into by-products. Although the Cu can interact with Ti active sites [20], it has little catalytic effect in the ammoximation of MEK. This may be explained that copper may catalyze $H_2O_2$ to produce highly active free radicals [23], effectively oxidizing and decomposing MEK and MEKO. In the ammoximation of MEK catalyzed by Fe-TS-1 and Co-TS-1, the conversion of MEK was still lower than that of TS-1, which may be attributed to the oxides of iron or cobalt that can decompose $H_2O_2$, which is not conducive to the ammoximation of MEK. For the introduction of Fe, the selectivity of MEKO slightly improved from 92.8% of TS-1 to 95.6%. It confirmed that the electronic effect of Fe on Ti centers would activate TS-1 activity [21]. When the rare earth element cerium was introduced, the conversion of MEK was 97.3%, which was similar to TS-1, probably because the cerium's atomic radius was too large to recombine with TS-1. It was worth noting that Ni-TS-1 displayed the best catalytic activity in the conversion of MEK to MEKO. Relative to TS-1, the selectivity of MEKO (99.3%) was significantly improved while maintaining high conversion of MEK (99%). Thus, Ni-TS-1 was intensively studied in the following research.

**Table 1.** Comparisons of the catalytic performance for the ammoximation of methyl ethyl ketone (MEK) over various catalysts.

| Catalyst | Conversion (%) | Selectivity (%) |
|---|---|---|
| TS-1 [a] | 98.9 | 92.8 |
| Fe-TS-1 [a] | 96.5 | 95.6 |
| Co-TS-1 [b] | 90.7 | 91.3 |
| Ni-TS-1 [a] | 99 | 99.3 |
| Cu-TS-1 [b] | 74.1 | 0 |
| Ce-TS-1 [a] | 97.3 | 90.6 |

[a] Reaction conditions: MEK, 0.1 mol; t-butanol, 25 mL; catalyst, 1.00 g; temperature, 343 K; total reaction time, 2 h. The $H_2O_2$ and $NH_3 \cdot H_2O$ were added at a constant rate for 1.5 h. [b] Total reaction time, 3.5 h. The $H_2O_2$ and $NH_3 \cdot H_2O$ were added at a constant rate for 1.5 h. The other conditions were the same with [a].

### 2.1.2. The Catalytic Activity of XNi-TS-1

In order to optimize the catalytic activity of Ni-TS-1, the effects of Ni dosage in Ni-TS-1 on the ammoximation of MEK was investigated. The results are shown in Table 2. The selectivity of MEKO went up and then down gradually with the increase of Ni dosage. When the dosage was 3 wt %, the catalytic activity was the best. This apparent change may be associated with the structure of the catalyst. When the Ni dosage was small, nickel species did not affect the Ti active sites. When the Ni dosage was too high, there were many Ni species to block the channels and cover Ti active sites, thus it was adverse to catalytic reactions.

**Table 2.** The effects of Ni dosage on the ammoximation of MEK over Ni-titanium silicalite-1 (TS-1).

| Catalyst | Conversion (%) | Selectivity (%) |
|---|---|---|
| 1 wt % Ni-TS-1 | 99.0 | 92.4 |
| 2 wt % Ni-TS-1 | 98.8 | 93.9 |
| 3 wt % Ni-TS-1 | 99.0 | 99.3 |
| 4 wt % Ni-TS-1 | 99.1 | 96.7 |
| 5 wt % Ni-TS-1 | 99.1 | 91.4 |

Reaction conditions: MEK, 0.1 mol; t-butanol, 25 mL; catalyst, 1.00 g; reaction temperature, 343 K; total reaction time, 2 h. The $H_2O_2$ and $NH_3 \cdot H_2O$ were added at a constant rate for 1.5 h.

### 2.2. Reusability Tests of 3% Ni-TS-1

The main advantages of catalysts in heterogeneous catalytic systems are its good stability and recyclability. The 3 wt % Ni-TS-1 catalyst was reused and its service life was tested. As can be seen from Figure 1, the conversion of MEK and the selectivity of MEKO were not affected when used

twice. When the catalyst was reused five times, the conversion of MEK remained above 90%, and the selectivity of MEKO was over 74.1%. The decrease of catalytic activity may be attributed to the loss of a small number of catalysts, increase of surface acidity, and the obstruction of active sites by organic species. The catalyst after each reuse was dried. Then thermogravimetric analysis was carried out, and the weight loss rate was calculated (Table 3). It was found that the weight loss rate increased with the number of uses, and the blockage of pore channels by organic substances became more and more serious. This may be the main factor affecting the catalytic activity. After using five times, 3 wt % Ni-TS-1 was regenerated by calcination and applied to MEK ammoximation reaction (Figure 1). The catalytic activity of 3 wt % Ni-TS-1 was completely restored.

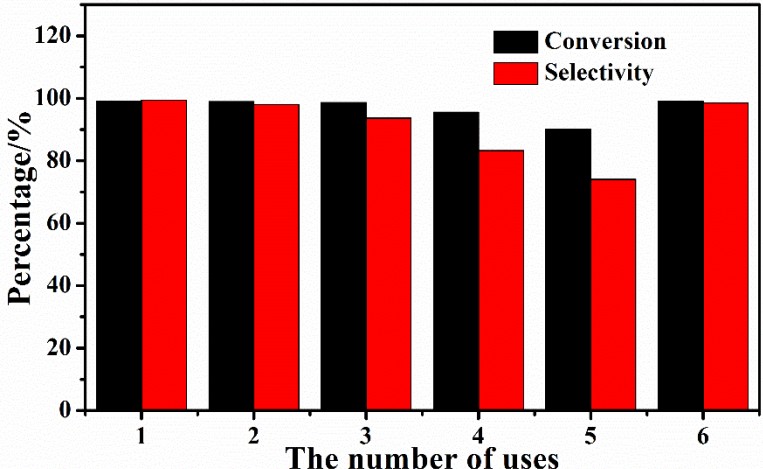

**Figure 1.** Repetition and regeneration of 3 wt % Ni-TS-1 on the ammoximation of MEK (Reaction conditions: MEK, 0.1 mol; t-butanol, 25 mL; catalyst, 1.00 g; reaction temperature, 343 K; total time, 2 h. The total time of other reactions were 3.5 h. The $H_2O_2$ and $NH_3 \cdot H_2O$ were added at a constant rate for 1.5 h).

**Table 3.** Thermogravimetric analysis after repeated use of 3 wt % Ni-TS-1.

| The Number of Uses | Weight Loss Rate (%) |
|---|---|
| 1 | 5.10661 |
| 2 | 8.21042 |
| 3 | 8.90195 |
| 4 | 9.68479 |
| 5 | 10.20894 |

## 2.3. Characterization of Catalysts

### 2.3.1. XRD and FT-IR Characterization

The crystal structure information of commercial TS-1, M-TS-1, and XNi-TS-1 were analyzed by the XRD (Figure 2A,B). All samples exhibited same diffraction peaks at 2θ = 7.8°, 8.8°, 23.2°, 23.8°, and 24.3°, which represented that the MFI skeleton structure of TS-1 was not changed or destroyed after metal modification. There was no characteristic diffraction peak of metal oxides in the XRD patterns. It may be that parts of metal entered the framework of TS-1 and a small amount of metal oxides were well dispersed on the TS-1 surface by the ultrasonic impregnation synthesis method [24].

The stretching vibration of chemical bonds or functional groups in the TS-1 and M-TS-1 samples were detected by FT-IR spectroscopy (Figure 2C). The vibration peaks at 550 cm$^{-1}$ and 1225 cm$^{-1}$ were characteristic peaks of the molecular sieve with MFI topology; 450 cm$^{-1}$ was the bending vibration peak of Si–O bond; 1110 cm$^{-1}$ and 800 cm$^{-1}$ absorption peaks corresponded to antisymmetric and symmetrical stretching vibration of internal silicon-oxygen tetrahedral units [25]; the characteristic

peak of 960 cm$^{-1}$ indicated the existence of the tetra-coordinated framework titanium. After the metals modified TS-1, the characteristic peak of 960 cm$^{-1}$ still existed obviously. However, the modification of metals made the infrared absorption band shift at around 966 cm$^{-1}$. In the different metals used to modify the TS-1 catalyst, the metal with the greatest wave number migration is Ni, followed by Fe, Co, and Ce. With the addition of Cu, wave number migration is almost not realized. The local FT-IR spectrum of XNi-TS-1 is shown in Figure 2D. It can be seen that the introduction of Ni caused a deviation in the IR absorption bands around 965 cm$^{-1}$ to lower wave numbers. The wave number of [Ti–O] bond vibration decreased from 965.37 cm$^{-1}$ to 962.11 cm$^{-1}$ as the dosage of Ni increased from 1 wt % to 5 wt %, which indicated that the introduction of Ni weakened the [Ti–O] bond. It may be related to the charge transfer of [Ti–O] in the tetrahedral [26]. In other words, the interactions between Ni and Ti decreased the strength of the titanium oxygen bond and increased the electrophilicity of the Ti center as the dosage of Ni in the TS-1 increased, which made a substantial contribution to the satisfactory conversion of MEK to MEKO.

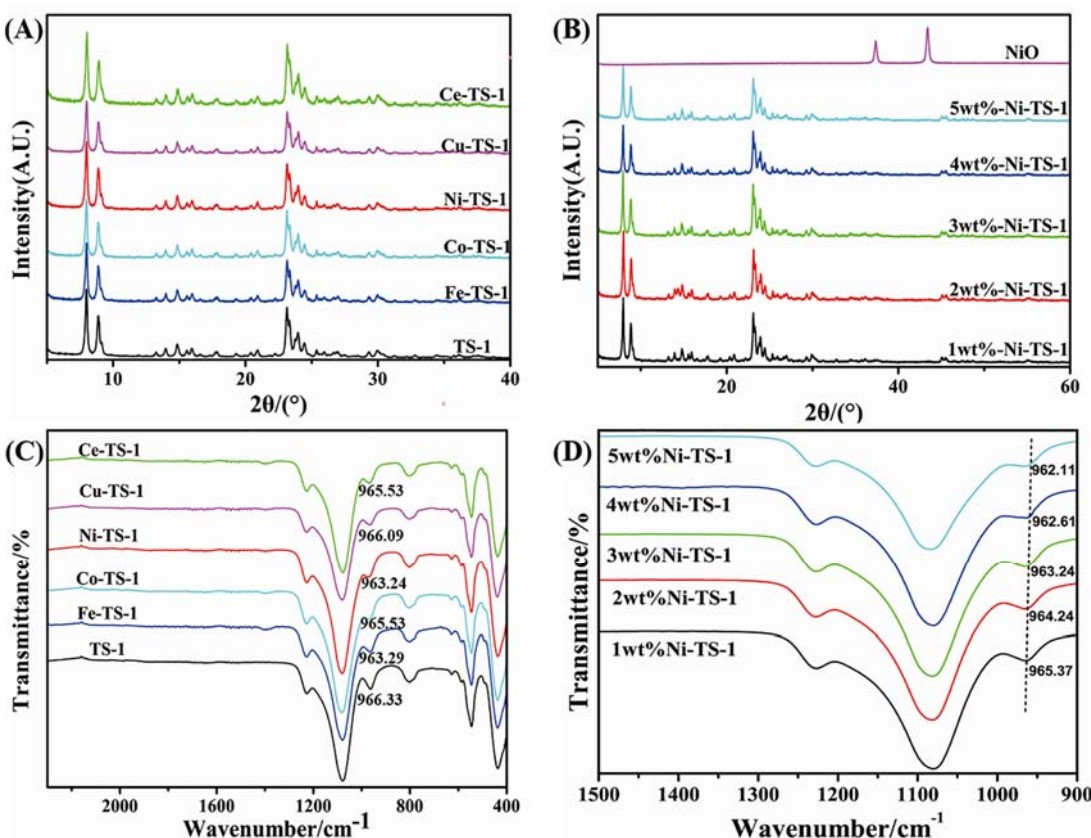

**Figure 2.** (**A**) XRD patterns of M-TS-1, (**B**) XRD patterns of XNi-TS-1, (**C**) FI-IR spectra of M-TS-1, and (**D**) FI-IR spectra of XNi-TS-1.

### 2.3.2. SEM Characterization

Microstructure of samples was further monitored by SEM. As can be seen from Figure 3, both TS-1 and Ni-TS-1 samples showed uniform spherical particles with the size of about 0.1–0.2 μm. Compared with TS-1, the size and shape of Ni-TS-1 did not have an obvious change. These results showed that the metal oxide did not destroy the crystal structure of TS-1, nor did it change the morphology of TS-1 particles.

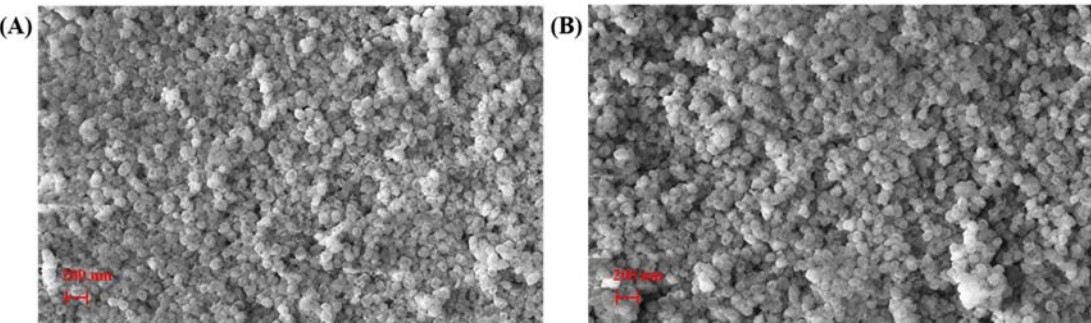

**Figure 3.** SEM images of TS-1 (**A**) and 3 wt % Ni-TS-1 (**B**).

### 2.3.3. Induced Coupled Plasma-Atomic Emission Spectroscopy (ICP-AES) and $N_2$ Adsorption-Desorption Characterization

The results of ICP-AES for the XNi-TS-1 catalyst are shown in Table 4. The loading efficiency (*e*) of Ni on TS-1 was about 70%.

**Table 4.** Ni content, specific surface area, and micropore volume of TS-1 and XNi-TS-1.

| Catalyst | $w$ (Ni)(%) Actual | $e$ (Ni)(%) | BET Surface area/$m^2 \cdot g^{-1}$ | Micropore Volume/$cm^3 \cdot g^{-1}$ |
|---|---|---|---|---|
| TS-1 | - | - | 394.7 | 0.0722 |
| 1 wt % Ni-TS-1 | 0.73 | 73.0 | 388.0 | 0.0719 |
| 2 wt % Ni-TS-1 | 1.39 | 69.5 | 387.9 | 0.0714 |
| 3 wt % Ni-TS-1 | 2.07 | 69.0 | 384.9 | 0.0709 |
| 4 wt % Ni-TS-1 | 2.75 | 68.8 | 377.9 | 0.0708 |
| 5 wt % Ni-TS-1 | 3.43 | 68.6 | 363.3 | 0.0699 |

$N_2$ adsorption–desorption isotherm of TS-1 and XNi-TS-1 are shown in Figure 4. According to the BDDT classification, all of the samples showed type IV isotherms with type H3 hysteresis loop, indicating the presence of mesopores. In addition, the introduction of Ni did not affect the presence of mesopores in TS-1. As shown in Table 4, the decrease in surface area and pore volume can be observed with the increase of Ni dosage. The $N_2$ adsorption capacity of the Ni-TS-1 decreased compared to TS-1. This suggested that the metal oxides partially occupied the microporous channels.

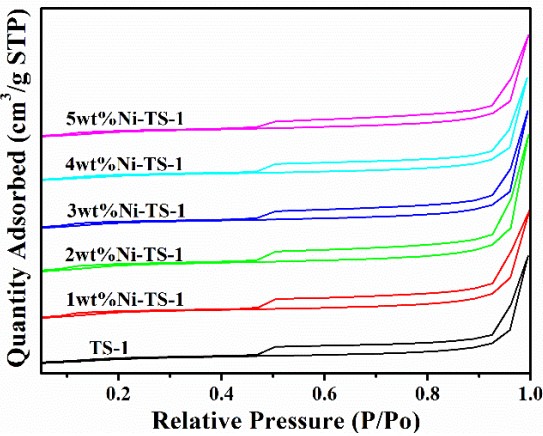

**Figure 4.** $N_2$ adsorption–desorption isotherm of TS-1 and XNi-TS-1.

### 2.3.4. DR UV-Vis Characterization

DR UV-Vis spectroscopy is an effective way to understand the nature of active Ti species on TS-1. The DR UV−Vis spectroscopy results of TS-1 and XNi-TS-1 are presented in Figure 5. In the DR UV-Vis

spectra, the absorption peak near 210 nm indicated the existence of tetra-coordinated skeleton titanium, which was originated from the electronic transfer of the p$\pi$–p$\pi$ transitions between titanium and oxygen in the framework titanium species. The absorption peaks around 330 nm was anatase $TiO_2$ outside of the framework [27–29]. It can be seen from Figure 5 that the absorption peak of TS-1 near 330 nm was large, and the peak of Ni-TS-1 still existed, but there was a tendency to become smaller from 1 wt % Ni-TS-1 to 5 wt % Ni-TS-1. This may be due to the addition of Ni, which caused some non-skeletal titanium to be lost. In addition, no characteristic band at around 362 nm is observed in XNi-TS-1 samples, which is assigned to NiO. This conclusion agreed with the XRD results.

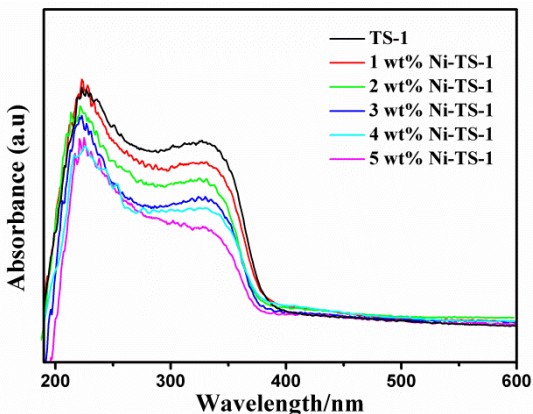

**Figure 5.** DR UV-Vis spectra of TS-1 and XNi-TS-1.

### 2.3.5. XPS Characterization

The surface chemical status and the interaction between the nickel component and TS-1 were investigated by XPS. Figure 6A shows the XPS full spectra of 3 wt % Ni-TS-1. Obviously, 3 wt % Ni-TS-1 had peaks of titanium, nickel, and silicon. The spectra of Ni 2p in 3 wt % Ni-TS-1 were fitted into four peaks in Figure 6B. The symbolic peaks at 856.5 eV and 874.2 eV are attributed to the binding energies of Ni $2p_{3/2}$ and Ni $2p_{1/2}$, which were assigned to $Ni^{2+}$. The Ni $2p_{3/2}$ XPS spectrum of free NiO usually shows peaks at 855.6 eV [30], while the Ni $2p_{3/2}$ XPS spectrum of 3 wt % Ni-TS-1 showed peaks at 856.5 eV. The increase of binding energy values indicated that Ni species in modified samples was afforded electrons by TS-1. The two satellite peaks, at a binding energy of 862.3 eV and 880.1 eV, for $Ni^{2+}$ were also observed [31]. Figure 6C shows the Ti 2p spectra of TS-1 and 3 wt % TS-1. The two characteristic peaks located at 460.2 eV and 465.2 eV were attributed to Ti $2p_{3/2}$ and Ti $2p_{1/2}$, respectively [20]. The peak of Ti $2p_{3/2}$ could be decomposed into two components. The peaks at 458.7 eV and 460.2 eV were assigned to anatase $TiO_2$ and framework Ti species, respectively [32]. For nickel-modified TS-1 catalyst, in addition to the characteristic peak at 460.5 eV, another characteristic peak appeared at 459.7 eV. The decrease of binding energy may be due to the migration of Ti 2p orbital electron cloud in the skeleton, which reduced the density of the electron cloud around the Ti center. Combined with Ni 2p XPS results, the addition of Ni reduced the density of electron cloud around the Ti center in TS-1, and enhanced the electrophilicity of the Ti center, which can improve the catalytic activity of TS-1 on the selective catalytic oxidation reaction.

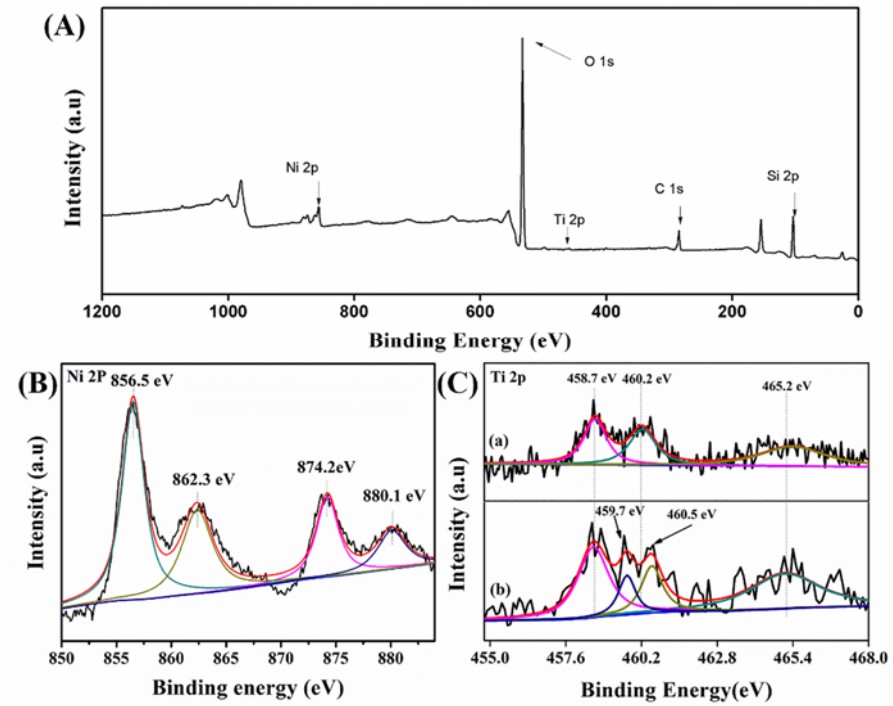

**Figure 6.** (**A**) XPS full spectra of 3 wt % Ni-TS-1, (**B**) Ni 2p, and (**C**) Ti 2p XPS of (**a**) TS-1and (**b**) 3 wt % Ni-TS-1.

### 2.3.6. The Analysis of Point of Zero Charge

Point of zero charge (PZC) is the pH when the net charge on the solid surface is zero in the aqueous solution. It is an important parameter for calibrating acidity and basicity of a solid surface [33]. The PZC of TS-1 and 3 wt % Ni-TS-1 is shown in Figure 7. The PZC of TS-1 and 3 wt % Ni-TS-1 was 3.00 and 6.25, respectively. When TS-1 was modified by nickel, the PZC of the catalyst was raised. This striking observation could be closely related to the structure and characteristic of the sample. Firstly, Ni replaced Si into the framework of TS-1, because the PZC of NiO and $SiO_2$ is 8.33 and 3.00, respectively [34,35]. In addition, the acidic sites on the surface were covered by nickel species. The rise of PZC was beneficial to maintaining higher pH of ammoximation system of MEK and inhibiting the further oxidation of MEKO.

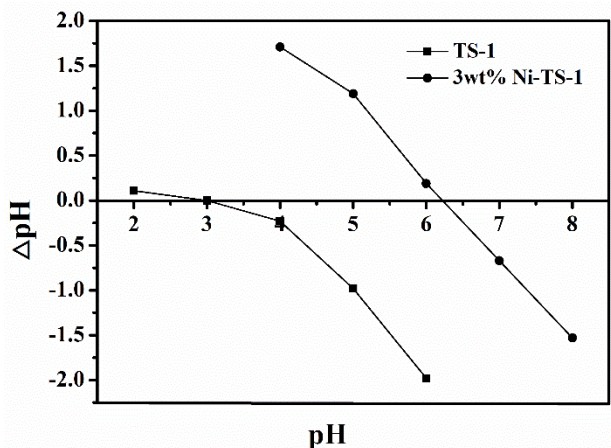

**Figure 7.** The point of zero charge (PZC) of TS-1 and 3 wt % Ni-TS-1.

### 2.4. Mechanism Analysis

The results of characterization illustrated parts of modified Ni were incorporated into the TS-1 skeleton as replacements of some Si sites and interacted with Ti active site and improved the electrophilicity of the Ti center [26], which is beneficial to adsorb $H_2O_2$ in the reaction system. the mechanism of liquid-phase ketoammoximation is hydroxylamine mechanism [36]. According to literature reports and our experimental results, we proposed the mechanism of Ni-TS-1 catalyzed the ammoximation of MEK (Scheme 2). Firstly, the four-coordinated titanium species combines with $NH_3 \cdot H_2O$ to become the six-coordinated state, further forming titanium peroxide under the action of $H_2O_2$. Then titanium peroxide reacts with $NH_3 \cdot H_2O$ to form ammonium peroxide, which would easily release $NH_2OH$. Eventually, $NH_2OH$ would react with MEK to produce MEKO by non-catalytic oxidation [37]. The addition of Ni reduced the electron cloud density of the titanium active center, improved its electrophilicity, enhanced the adsorption capacity for $H_2O_2$, and thus increased the reaction rate of the control step of ketoammoximation, i.e., the formation rate of hydroxylamine. In addition, the increase of basicity of the Ni-modified catalyst was beneficial to controlling the alkaline environment of the reaction system and effectively inhibiting the occurrence of side reactions. Therefore, Ni-TS-1 exhibited a good catalytic effect in the ammoximation of MEK.

**Scheme 2.** Reaction mechanism of ammoximation of MEK catalyzed by Ni-TS-1.

## 3. Experimental Section

### 3.1. Materials

The industrial catalyst TS-1 ($SiO_2/TiO_2 = 56$, specific surface area of 394.7 $m^2/g$) was provided by Luxi Chemical Group Co., Ltd. Analytical grade ammonia (25%, AR, Laiyang, Shandong, China), hydrogen peroxide (30%, AR, Laiyang, Shandong, China), t-butanol (≥99%, AR, Shanghai, China), MEK (≥99%, AR, local vendor), nickel nitrate hexahydrate (98%, AR, Shanghai, China), Cobalt nitrate hexahydrate (98%, AR, Shanghai, China), Ferric nitrate nine-hydrate (≥99%, AR, Tianjin, China), Copper nitrate trihydrate (≥99%, AR, Tianjin, China), Cerium nitrate hexahydrate (98%, AR, Shanghai, China), and Anhydrous sodium carbonate (≥99.8%, AR, Xuzhou NO.2 Reagent Factory) were all obtained commercially.

### 3.2. Methods

### 3.2.1. Catalyst Preparation

The transition metal-modified TS-1 was prepared by an ultrasonic impregnation method [20]. A total of 4.00 g TS-1 was dispersed in 20 mL of deionized water, and sodium carbonate ($Na_2CO_3:M^{n+} = 2:1$) was added as precipitator. Under vigorous stirring, the corresponding aqueous solution of metal nitrate with certain quality was slowly dripped. The mixture was sonicated for 1 h and stirred

vigorously for 3 h. Then the mixture was filtered and the solid was dried at 120 °C overnight and calcined at 550 °C for 6 h to obtain M-TS-1 (M is Fe, Co, Ni, Cu, Ce). The loading X is defined as m (metal element)/m (TS-1), and the range of X is 1–5 wt %.

### 3.2.2. Catalyst Characterization

Powder X-ray diffraction (XRD) was performed on a Rigaku Smartlab SE X-ray diffractometer with a Cu-kα radiation source at a tube voltage of 40 kV and a tube current of 25 mA. Diffraction patterns were recorded between 5° and 60°. FI-IR spectroscopy was measured on a Nicolet 380 spectrometer over a range of 4000 $cm^{-1}$–400 $cm^{-1}$ at a resolution of 4 $cm^{-1}$. $N_2$ adsorption–desorption isotherms were collected by a Gemini VII analyzer. Prior to the tests, about 100 mg samples were degassed at 250 °C for 4 hours. Then, the sample completing the pretreatment was subjected to a nitrogen adsorption–desorption isotherm test at a liquid nitrogen temperature (77 K). DR UV-Vis was performed on a TU-1901 spectrophotometer in the wavelength range from 800 to 190 nm. The induced coupled plasma-atomic emission spectroscopy (ICP-AES) was carried out using a 1000-type spectroscopy. The sample of the catalyst is completely dissolved by 40% HF, and the sample was made up to 50 mL with water after HF removal by electric heating plate. The metal content in the catalyst is quantitatively detected. The loading efficiency (*e*) of Ni on TS-1 was calculated based on the actual loading of Ni element measured and the amount of Ni in the impregnating solution. TGA measurements were rendered on a DTG60A thermogravimetric analyzer from room temperature to 800 °C at the rate of 10 °C/min. The weight loss rate was the ratio of weight loss mass of catalyst to original mass of catalyst. The X-ray photoelectron spectroscopy was collected by A ESCALAB250 spectroscopy. Scanning electron microscopy (SEM) was performed on a JSM-6700F cold field high-resolution emission scanning electron microscope.

### 3.2.3. Point of Zero Charge (PZC) of Catalyst

The point of zero charge of catalyst ($pH_{PZC}$) was determined by the salt titration method: 0.4 mol/L $NaNO_3$ solution was prepared. We took 20 mL $NaNO_3$ solution and adjusted the pH to 2, 3, 4, 5, 6, 7, and 8 with NaOH or $H_2SO_4$. Then, 0.2 g catalyst was added to shake for 0.5 h until the pH was stabilized, and the pH change value (△pH) before and after the addition of catalyst was calculated. The initial pH was horizontal coordinate and △pH was vertical coordinate. When the ordinate is zero, the pH value is PZC.

### *3.3. Catalytic Activity Test*

A total of 0.1 mol MEK, 1.00 g catalyst, and 25 mL t-butanol were placed in a four-neck round bottom flask with a condensation reflux device. The constant temperature water bath was used to keep the temperature of the reaction system at 343K. $NH_3·H_2O$ and $H_2O_2$ were slowly dropped into the flask for 1.5 h (MEK:$NH_3·H_2O$:$H_2O_2$ = 1:4:1.1), and then the reaction lasted for a certain period of time after the end of dropping. After the reaction, the catalysts were filtrated and separated. The content of MEK (standard curve method) was detected by high performance liquid chromatography (HPLC). The content of MEKO (standard curve method) was detected by gas chromatography (GC) after the filtrate was volumed with absolute ethanol.

The 3 wt % Ni-TS-1 catalyst was reused. The catalyst, for the first reaction, was completely transferred to the second reaction, and so on. The catalyst was reused for five times with other conditions unchanged. In addition, the catalyst that was reused five times was calcined (540 °C for 7 h) to activate and regenerate, and applied to the ammoximation of MEK again.

$$\text{Conversion of MEK (\%)} = \frac{\text{Converted moles of MEK}}{\text{Initial moles of MEK}} \times 100\%, \quad (1)$$

$$\text{Selectivity of MEKO (\%)} = \frac{\text{Moles of MEKO}}{\text{Converted moles of MEK}} \times 100\%. \quad (2)$$

Liquid chromatography conditions: Shimadzu HPLC LC-20AT; the column is Diamons C18 column (250 mm × 4.6 mm, 5 μm), the UV detection wavelength is 274 nm, the mobile phase is V $_{methanol}$:V $_{water}$ = 50:50, and the flow rate is 0.8 mL/min.

Gas chromatographic conditions: Shimadzu gas chromatograph GC-2010; the column is FFAP capillary column, the carrier gas is high purity nitrogen, the inlet and the detection port temperature is 200 °C, and the column temperature is 100 °C.

## 4. Conclusions

Five kinds of transition metal-modified TS-1 catalysts were prepared and employed to catalyze the ammoximation of MEK. The catalytic activity of various transition metal-modified TS-1 was diverse. Compared with TS-1 and other M-TS-1, 3 wt % Ni-TS-1 showed the best catalytic activity. The conversion of MEK and the selectivity of MEKO reached 99.0% and 99.3%, respectively. Furthermore, the catalyst was fully recovered without considerable decrease of MEK conversion and MEKO selectivity. The characterization of Ni-TS-1 showed that the introduction of nickel species did not change the skeleton structure of TS-1. Meanwhile, the nickel entering the framework of TS-1 effectively changed the electron environment of the Ti active sites and increased the electrophilicity of TS-1. In addition, the surface nickel species reduced the surface acidity of the catalyst, inhibiting the deep oxidation of the MEKO. We believe the obtained metal-modified TS-1 catalyst is very promising for the ammoximation of MEK and relative reactions for industrial applications.

**Author Contributions:** S.J. and X.X. conceived and designed the experiments and guided the research. D.Y., H.W., W.W., S.P., and X.Y. performed characterization and catalytic studies. All authors analyzed and discussed the results. D.Y., S.J., and X.X. analyzed the data and wrote the paper.

**Funding:** This research received no external funding.

**Acknowledgments:** This research was financially supported by the science and technology development plan of Shandong Province (2013GZX20109)**.**

**Conflicts of Interest:** The authors declare no conflict of interest.

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
