# Peer review of "Nickel-Modified TS-1 Catalyzed the Ammoximation of Methyl Ethyl Ketone"

_catalysts, doi:10.3390/catal9121027_

Round 1
Reviewer 1 Report
The manuscript titled "Nickel-modified TS-1 catalyzed the ammoximation of methyl ethyl ketone" is related to the application of Me-TS-1 to oximate MEK the findings can be of interest to the wider community investigating. The manuscript is very well written but it needs some improvements as listed below:
Comment 1: The introduction section must be improved using newer references. For instance there are recent works about oximation of cyclohexanone can be used to show the interest of the oximes at industrial scale.
Comment 2:Have the experimental conditions (T, time..., reagent concentrations) been selected attending to the industrial conditions?
Comment 3: In Figure 1, use number 6 is after the regeneration of the catalyst, It might be helpful add this information in the figure caption.
Comment 4: It is not very clear in the text, which is the main scope of the ICP analysis, is it to study the leaches of the metal? the total metal content in the catalyst surface? How was this analysis accomplised?.
Comment 5:Scheme 2, are evidences of this mechanism in literature?
Comment 6: The experimental procedure of the oximation reaction must be improved. How was the H2O2 added? How was achieves constant temperature?
Comment 7: Have the impurities generated during the ammoximation been indentified?
Reviewer 2 Report
Catalytic character of Ni-modified TS-1 for ammoximation of methyl ethyl ketone has been described.
1) Application for several other ketones with Ni-modified TS-1 will be described
2) It is slightly difficult to make reproducibility between Footnote in Table 1 and 2 and Experimental section.
I) Footnote in Table 1 and 2
Reaction conditions: MEK, 0.1 mol; t-butanol, 25 mL; catalyst, 1.00 g; reaction temperature, 343 K; reaction time, 2 h. The H2O2 and NH3•H2O were added at a constant rate for 1.5 h.
II) lane 289 3.3 Catalytic Activity Test
0.1 mol MEK, 1.00 g catalyst and 25 mL t-butanol were placed in a three-neck round bottom flask with condensation reflux device, and NH3•H2O and H2O2 were slowly added during the reaction process (MEK:NH3•H2O:H2O2=1:4:1.1).
Did "0.4 mol of NH3•H2O and 0.11 mol H2O2" add to the reaction mixture over 90 min?
What did the reaction time "2 h" mean? (addition over 1.5h + reaction for 0.5h? or addition over 1.5 h and reaction for 2 h?)
3) Experiments with less amounts of NH3•H2O has been conducted to ensure the proposed mechanism.
Round 2
Reviewer 2 Report
Thank you for preparing the revised manuscript.
And reviewer can agree to publish the manuscript on this journal.